# Bee Guilds' Responses to Urbanization in Neotropics: A Case Study †

**Sônia Guimarães Alves** *  **and Maria Cristina Gaglianone**

Laboratório de Ciências Ambientais, PPG-Ecologia e Recursos Naturais, Universidade Estadual do Norte Fluminense Darcy Ribeiro, CBB, Avenida Alberto Lamego 2000, Campos dos Goytacazes 28013-602, Brazil; mcrisgag@uenf.br
* Correspondence: soniagui68@gmail.com; Tel.: +55-(22)-998-001-578
† This paper is an extended version of the paper published in the 1st International Electronic Conference on Biological Diversity, Ecology, and Evolution (BDEE 2021), Online, 15–31 March 2021.

**Abstract:** The consequent deforestation of urban sprawl is one of the causes of the decline of wild bee communities. In this context, urban green areas (UGA) may play an important role and constitute refuge areas for bees. This study analyzed the influence of UGA conditions and their surroundings in bee guilds' responses in a medium-sized Brazilian city (Campos dos Goytacazes, RJ). The bees were sampled for 12 months (2017–2018) in 12 UGAs, and bee abundance and species richness were evaluated in guilds considering: nesting behavior, nesting site, and trophic specialization. We used as explanatory variables conditions of UGAs—the number of trees (NT), diameter at breast height (DBH), flower cover (FC), plant richness (PR), percentage of paving (PV)—and of their surroundings— paving (SPV) and the number of buildings (NB). Results showed 80% of eusocial bees, 82% nest in cavities, and 99% were generalists. FC, DBH, and NB mainly explained the responses of different guilds in study areas from all explanatory variables. Thus, this study confirms different responses associated with bee guilds' attributes. In order to conserve bee diversity, city planning must include more green areas with large flower covers and avoid long corridors of high buildings that can impact bee dispersion.

**Keywords:** bees; conservation; pollinators; urban management

## 1. Introduction

Cities in the world have faced steady growth in the last decades caused by migration from rural areas. As a result, the urban population is around 55%, and the UN estimates that the world urban population might increase by over 68% by the year 2050, which will demand faster urban expansion. The changes resulting from urbanization will affect mainly medium-sized cities (between 500 thousand and one million inhabitants), where half of the world's urban population currently lives [1]. Usually, the urbanization process in tropical and developing countries is not planned and causes drastic changes in the landscape, often irreversible, such as the increase of impervious areas (pavement, asphalt, buildings) and the destruction of wild vegetation. The great challenge for these cities is the sustainable growth that guarantees housing, transport system and energy for the population, combined with environment preservation and conservation of biodiversity [2].

The expansion of urban areas is one of the leading causes of pollinators' decline [3–5]. Among the affected pollinators are the wild bee species (Hymenoptera, Apoidea). Many bee species have a short lifecycle and high sensitivity to temperature, luminosity and humidity, responding to changes caused by urbanization [6]. Bees vary in their nesting location, build nests in the ground, in preexisting cavities in hollows in tree trunks and branches, human constructions such as walls and poles, or even exposed, out of cavities [7]. Bee species also differ in the degree of sociality, ranging from solitary to highly eusocial, and in the degree of specialization or generalization in the choice of resources [8]. Therefore,

the bee community suffers direct influence from the availability of plant resources in the environment. Vegetation offers substrates such as trunks and hollows of trees for nesting, and their flowers provides resources such as nectar, pollen, oils and resin for feeding and provisioning the nests [7,9]. These different bees' biological characteristics determine how these insects interact with the environment and how they tolerate changes [10].

On the other hand, urban green areas (UGAs) such as squares, parks, urban forests and gardens are growing in importance, considering bee-friendly spaces with potential to act as a refuge for bees [11]. Urban green areas are patches of vegetation within cities, where there is a lower application of pesticides when compared to agricultural areas. The vegetation in these urban areas is composed of wild or exotic plants inserted through landscaping projects that can result in a more heterogeneous environment. In addition, these green areas' characteristics can provide nesting sites, foraging and sources of other materials for bees [12]. Therefore, even small green areas have importance and can function as ecological corridors, connecting patches of vegetation to improve bee distribution [13].

However, there is still no consensus on which characteristics of urban green areas are essential to conserving the bee community in cities [11]. One expects that pollinators respond to urbanization with positive and negative effects observed depending on taxon and environment traits. Urban expansion is not homogenous and has a significant variation in different countries and ecosystems [14]. The available data are not representative enough to guide pollinators' conservation actions globally, because most focus on North America and Western Europe with specific taxonomic groups and geographic situations. The results cannot be applied in other regions or taxa [15]. A recent systematic review of 141 peer-reviewed journals, that aimed to understand how urbanization affects pollinators' communities, showed that tropical regions remain little studied [16]. It is a significant gap that needs to be fulfilled, considering that these regions have incredible biodiversity, and many places are hotspots threatened by fast urban growth. South American countries already have an urban population of over 80%, and Brazil is at the top of the list, with more than 200 million people living in cities [1].

Here, we used a medium-sized Brazilian city to investigate which conditions of urban green areas and their surroundings can potentially contribute to shelter and conserve bees' communities. Considering biological and functional traits, we also grouped bees in guilds to provide data that can increase the efficiency of management and conservation of bee and plant species [17]. Therefore, this work aims to answer the following questions: (1) What is the community structure found in urban green areas associated with a medium-sized city in Brazil? (2) How do the different bee guilds respond to the conditions of urban green areas? Answering these questions, we intend to test two hypotheses. First, urban green areas can provide nesting sites and foraging resources to provide refuge for the bees' community. Second, bee species respond to urbanization according to their nesting behavior, nest site and trophic specialization.

## 2. Materials and Methods

### 2.1. Study Area

This study was conducted in Campos dos Goytacazes, RJ, Brazil, with approximately 4.037 km$^2$ and an urban area of 87.73 km$^2$. The total population is 507.548 thousand inhabitants, and almost 85% of the population lives in the urban area [18]. The climate is hot and humid tropical AW (Köppen-Geiger classification), with dry winter (April to September) and rainy summer (October to March), and average annual rainfall between 800 and 1200 mm [19]. The average temperature is around 26 °C in the hottest months and 19 °C in the coldest ones. Most of the vegetation found in urban areas results from public and private planting in specific places [20], modified over time. The selection of sampling points was made by marking over an aerial Google Earth image of Campos dos Goytacazes and validating the presence of vegetation in loco. Twelve urban green areas (UGAs—U1 to U12) were selected, including squares, parks and gardens. ArcGIS 10.0 was used to create location map (Figure 1).

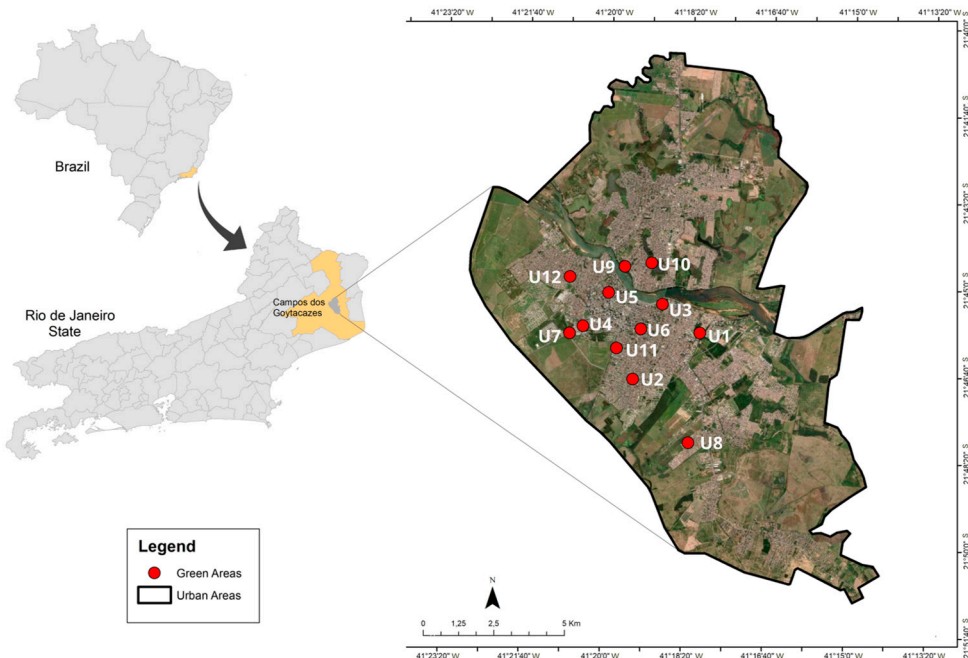

**Figure 1.** Red dots show the location of urban green areas (U1 to U12) evaluated in this study in the urban area of Campos dos Goytacazes, RJ, Brazil (Google image of the ArcGIS database).

*2.2. Bee Sampling*

We sampled the bees during their visits on flowers with an entomological net in the 12 UGAs between October 2017 and September 2018, with two samples in the rainy season and two samples in the dry season at each UGA. A sampling at each site was carried out by two collectors, who walked along the 12 UGAs searching for flowers in 3 periods of about 1 h each, between 7 am and 1 pm. Bee species were identified and deposited at the UENF Pollinating Insects Collection. They were classified by: (1) nesting site (soil or cavity), (2) nesting behavior (solitary, intermediate, eusocial) and (3) trophic specialization (generalist—collecting resources on flowers of a variety of plant species, specialist—collecting exclusively at a single plant genus). Most biological information derives from the literature [21,22], while for bee species whose functional traits were not available in the literature, we used information known for closely related taxa.

*2.3. Environmental Conditions*

In order to determine which environmental variables, influence the total richness and abundance of species, we measured the following traits in each study point: paved area, the richness of plants visited by bees, flower coverage, number of trees and diameter at breast height (DBH), of trees with more than 90 cm in circumference. The identification of the plant species where the bees were captured was made with the help of specialized literature [23–25] and the Flora Brazil project [26]. We measured the paved area inside the UGAs using the Google Earth Pro polygon tool (version 7.3.3.7786).

We identified the plants using photographs and exsiccates made with plant samples and deposited in Herbarium UENF. First, plant species inside the study areas were classified based on the following categories: native or exotic in Brazil, habit (herbaceous, shrubby, arboreal), nectar, pollen and/or oil as resources for pollinators [26,27]. Next, we estimated flower coverage for all flowering species on the same days of bee sampling. Finally, we calculated the area occupied by each plant with a measuring tape. From this total, we estimated the percentage covered by the flowers for each plant and we multiplied this percentage by the total plant coverage to obtain the flower coverage in square meters, we summed the results for all plants to obtain a day value and we considered the flower coverage in each UGA as the mean value for the four sampling days.

To analyze the environmental variables around the UGAs, we used the paved area around and the number of buildings with more than three floors that potentially can act as a barrier to bee dispersal, increasing isolation [28]. We measured these variables within a 500 m radius (buffers) from the center of each of the 12 UGAs. The paved area inside each buffer was measured with the polygon tool and we counted the buildings with more than three floors with the street view tool (both tools from Google Earth Pro, version 7.3.3.7786).

### 2.4. Data Analysis

To assess species diversity in each UGA, we calculated diversity indices of Shannon–Wiener (H′) and dominance of Berger–Parker [29]. In addition, we calculated the similarity between UGAs concerning the species composition fauna through the Bray-Curtis index.

The correlation between plant habit and species richness for each guild (related to trophic specialization, nesting behavior and nesting site) was evaluated using a Principal Component Analysis (PCA). These analyses underwent the Past 3.2 program, assuming a 95% significance level [30].

We constructed multivariate generalized linear models (GLMs) with negative binomial distribution to identify the effects of environmental variables (predictors: paved area, DBH of trees, plant richness, flower coverage, paved area around and number of buildings with more than three floors) on richness and abundance of bees in each guild. We tested the collinearity between the predictor variables using the variance inflation factor (VIF) of the car package of the R program. The best model was selected using the lowest value of the Akaike information criterion (AIC). We used the R version 4.0.5 program for these analyses, assuming a 95% significance level [31].

## 3. Results

### 3.1. Bee Community

In total, we collected 1163 bees of 39 species. Apidae was the family with the most extraordinary richness (19 species) and abundance (991 individuals). The tribes with the most remarkable species richness were Augochlorini (13) and Meliponini (5). The most abundant species were *Apis mellifera* (32% of individuals sampled), *Trigona spinipes* (26%) and *Plebeia droryana* (13%), all eusocial species. Among the non-eusocial species, the most abundant were Dialictus sp1 (4%), *Augochlora thalia* (3%) and *Xylocopa frontalis* (2%). The most remarkable species richness was found in the U1 area ($n = 20$) and the highest diversity indexes were found in U2 and U12 (H′ = 2.065 and H′ = 2.032), respectively. The highest dominance was registered in the U7 area (Berger–Parker = 0.7206) (Table 1).

The most similar areas were U7 and U8 (67%), and least similar were U7 and U4 (12%). The values of bee abundance and total species richness of sampled bees were higher in the rainy season (668 individuals and 31 species, respectively), when the average daily temperatures varied between 25 and 39 °C. In the dry season, with average daily temperatures between 18 and 38 °C, the abundance of bees was 495 and the richness was 28 species. Grouping bees in guilds, from all sampled individuals, 80% showed eusocial behavior, 11% belonged to a solitary bee and 9% had an intermediate level of sociality. Cavity bees represented 82% of the bees collected and 18% of soil-nesting bees. Considering trophic specialization, 99% of bee species were generalists. A higher richness of eusocial bees was found in the U5 and U6 areas, while intermediate and solitary species had a higher richness in the U1 area (Figure 2a,b). Solitary and soil-nesting bees composed the highest richness of sampled species and no eusocial/soil bee species were found (Figure 2c).

**Table 1.** Abundance of bee species collected on flowers with an entomological net at 12 Urban Green Areas (UGA) in Campos dos Goytacazes, RJ, Brazil, and respective biological characteristics: Nesting Behavior (NB)—Solitary (S), Eusocial (E), Intermediate (I); Nesting Site (NS)—Soil (SO), Cavity (C); Trophic Specialization (TS)—Generalist (G), Specialist (S).

| Species | Urban Green Area (UGA) | | | | | | | | | | | | Abundance | | Bee Guilds | | |
|---|---|---|---|---|---|---|---|---|---|---|---|---|---|---|---|---|---|
| | U1 | U2 | U3 | U4 | U5 | U6 | U7 | U8 | U9 | U10 | U11 | U12 | Total | Rel.% | NB | NS | TS |
| **APIDAE** | | | | | | | | | | | | | | | | | |
| **Apini** | | | | | | | | | | | | | | | | | |
| *Apis mellifera* L. | 64 | 11 | 19 | 8 | 12 | 24 | 98 | 56 | 22 | 46 | 15 | 6 | 381 | 32.7 | E | C | G |
| **Centridini** | | | | | | | | | | | | | | | | | |
| *Centris analis* Lep. | 2 | 2 | 0 | 0 | 0 | 1 | 0 | 0 | 0 | 0 | 0 | 0 | 5 | 0.4 | S | C | G |
| *Centris tarsata* Sm. | 1 | 0 | 0 | 0 | 0 | 0 | 0 | 1 | 0 | 3 | 1 | 0 | 6 | 0.5 | S | C | G |
| **Emphorini** | | | | | | | | | | | | | | | | | |
| *Melitoma segmentaria* (Fab.) | 9 | 0 | 0 | 0 | 0 | 0 | 0 | 0 | 0 | 0 | 0 | 0 | 9 | 0.8 | S | SO | S |
| **Euglossini** | | | | | | | | | | | | | | | | | |
| *Euglossa cordata* (L.) | 1 | 0 | 0 | 0 | 0 | 0 | 0 | 0 | 0 | 0 | 0 | 0 | 1 | 0.1 | I | C | G |
| *Euglossa* sp | 2 | 0 | 0 | 0 | 0 | 1 | 0 | 0 | 1 | 0 | 0 | 0 | 4 | 0.3 | I | C | G |
| *Eulaema flavescens* (Fr.) | 0 | 0 | 0 | 1 | 0 | 0 | 0 | 0 | 0 | 0 | 0 | 0 | 1 | 0.1 | I | SO | G |
| *Eulaema nigrita* Lep. | 0 | 1 | 0 | 0 | 0 | 2 | 0 | 0 | 0 | 0 | 0 | 0 | 3 | 0.3 | I | SO | G |
| **Exomalopsini** | | | | | | | | | | | | | | | | | |
| *Exomalopsis analis* Spinola | 4 | 0 | 0 | 0 | 0 | 0 | 2 | 0 | 1 | 0 | 0 | 0 | 7 | 0.6 | I | SO | G |
| *Exomalopsis auropilosa* Spinola | 0 | 0 | 0 | 0 | 0 | 1 | 5 | 1 | 0 | 0 | 0 | 0 | 7 | 0.6 | I | SO | G |
| **Meliponini** | | | | | | | | | | | | | | | | | |
| *Nannotrigona testaceicornis* (Lep.) | 0 | 0 | 0 | 0 | 52 | 0 | 0 | 0 | 0 | 0 | 12 | 0 | 64 | 5.5 | E | C | G |
| *Plebeia droryana* (Fr.) | 0 | 4 | 17 | 7 | 21 | 49 | 0 | 0 | 15 | 0 | 31 | 6 | 150 | 12.9 | E | C | G |
| *Plebeia* sp | 0 | 0 | 0 | 0 | 0 | 1 | 0 | 0 | 0 | 0 | 0 | 0 | 1 | 0.1 | E | C | G |
| *Tetragonisca angustula* (Latr.) | 0 | 15 | 0 | 0 | 1 | 0 | 0 | 0 | 0 | 0 | 0 | 0 | 16 | 1.4 | E | C | G |
| *Trigona spinipes* (Fab.) | 73 | 24 | 0 | 0 | 7 | 6 | 15 | 10 | 110 | 32 | 9 | 21 | 307 | 26.4 | E | C | G |
| **Xylocopini** | | | | | | | | | | | | | | | | | |
| *Xylocopa frontalis* Ol. | 0 | 2 | 0 | 0 | 4 | 2 | 0 | 0 | 9 | 0 | 1 | 2 | 20 | 1.7 | I | C | G |
| *Xylocopa nigrocincta* Brèthes | 1 | 0 | 0 | 0 | 0 | 0 | 1 | 0 | 0 | 2 | 0 | 0 | 4 | 0.3 | I | C | G |
| *Xylocopa ordinaria* Sm. | 0 | 2 | 1 | 0 | 0 | 0 | 0 | 0 | 0 | 0 | 0 | 0 | 3 | 0.3 | I | C | G |
| *Xylocopa suspecta* Moure & Camargo | 0 | 0 | 0 | 0 | 0 | 0 | 0 | 0 | 0 | 0 | 1 | 0 | 1 | 0.1 | I | C | G |
| **COLLETIDAE** | | | | | | | | | | | | | | | | | |
| **Hylaeini** | | | | | | | | | | | | | | | | | |
| *Hylaeus tricolor* (Schr.) | 0 | 0 | 0 | 0 | 0 | 1 | 0 | 0 | 0 | 5 | 0 | 0 | 6 | 0.5 | S | SO | G |
| *Colletes* sp | 0 | 0 | 0 | 0 | 0 | 0 | 0 | 0 | 0 | 0 | 1 | 0 | 1 | 0.1 | S | SO | G |

**Table 1.** *Cont.*

| Species | U1 | U2 | U3 | U4 | U5 | U6 | U7 | U8 | U9 | U10 | U11 | U12 | Total | Rel.% | NB | NS | TS |
|---|---|---|---|---|---|---|---|---|---|---|---|---|---|---|---|---|---|
| | | | | | Urban Green Area (UGA) | | | | | | | | Abundance | | Bee Guilds | | |
| **HALICTIDAE** | | | | | | | | | | | | | | | | | |
| **Augochlorini** | | | | | | | | | | | | | | | | | |
| *Augochlora (Oxytoglossella) thalia* Sm. | 4 | 5 | 11 | 2 | 0 | 0 | 0 | 4 | 0 | 2 | 4 | 2 | 34 | 2.9 | S | SO | G |
| *Augochlora (Augochlora) esox* Vachal | 3 | 2 | 0 | 1 | 0 | 0 | 0 | 4 | 0 | 0 | 0 | 2 | 12 | 1.0 | S | SO | G |
| *Augochlora* sp3 | 0 | 0 | 0 | 0 | 0 | 0 | 0 | 0 | 0 | 1 | 0 | 3 | 4 | 0.3 | S | SO | G |
| *Augochlora* sp4 | 1 | 1 | 0 | 1 | 0 | 0 | 0 | 0 | 2 | 0 | 2 | 0 | 7 | 0.6 | S | SO | G |
| *Augochlora* sp5 | 2 | 1 | 1 | 3 | 0 | 1 | 0 | 1 | 0 | 0 | 0 | 4 | 13 | 1.1 | S | SO | G |
| *Augochlora* sp6 | 1 | 0 | 0 | 0 | 0 | 0 | 0 | 0 | 0 | 0 | 0 | 0 | 1 | 0.1 | S | SO | G |
| *Augochlora* sp7 | 1 | 3 | 2 | 1 | 0 | 0 | 1 | 1 | 0 | 1 | 0 | 3 | 13 | 1.1 | S | SO | G |
| *Augochlora* sp8 | 3 | 0 | 0 | 0 | 0 | 1 | 1 | 2 | 0 | 0 | 0 | 2 | 9 | 0.8 | S | SO | G |
| *Augochlora* sp9 | 0 | 0 | 1 | 0 | 0 | 1 | 0 | 0 | 0 | 1 | 0 | 0 | 3 | 0.3 | S | SO | G |
| *Augochloropsis* sp1 | 0 | 0 | 0 | 0 | 2 | 0 | 0 | 0 | 0 | 2 | 0 | 0 | 4 | 0.3 | I | SO | G |
| *Augochloropsis* sp2 | 0 | 0 | 0 | 0 | 0 | 0 | 0 | 1 | 0 | 0 | 0 | 0 | 1 | 0.1 | I | SO | G |
| *Augochloropsis* sp3 | 1 | 0 | 0 | 0 | 0 | 0 | 0 | 0 | 0 | 0 | 0 | 0 | 1 | 0.1 | I | SO | G |
| *Augochloropsis* sp4 | 0 | 0 | 0 | 0 | 0 | 0 | 0 | 0 | 0 | 1 | 0 | 0 | 1 | 0.1 | I | SO | G |
| **Halictini** | | | | | | | | | | | | | | | | | |
| *Dialictus* sp1 | 5 | 1 | 1 | 1 | 0 | 1 | 13 | 19 | 4 | 2 | 0 | 4 | 51 | 4.4 | I | SO | G |
| **MEGACHILIDAE** | | | | | | | | | | | | | | | | | |
| **Megachilini** | | | | | | | | | | | | | | | | | |
| *Megachile affabilis* Mitchell | 0 | 0 | 0 | 0 | 0 | 0 | 0 | 0 | 0 | 0 | 1 | 0 | 1 | 0.1 | S | C | G |
| *Megachile neoxanthoptera* Cock. | 3 | 0 | 0 | 0 | 0 | 0 | 0 | 0 | 0 | 0 | 0 | 0 | 3 | 0.3 | S | C | G |
| **Anthidiini** | | | | | | | | | | | | | | | | | |
| *Dicranthidium seabrai* Urban | 0 | 0 | 0 | 0 | 0 | 0 | 0 | 0 | 0 | 0 | 2 | 0 | 2 | 0.2 | S | C | G |
| *Dicrantidium* sp | 2 | 0 | 0 | 0 | 0 | 0 | 0 | 0 | 0 | 4 | 0 | 0 | 6 | 0.5 | S | C | G |
| **Total sampled** | 183 | 74 | 53 | 25 | 99 | 92 | 136 | 100 | 164 | 102 | 80 | 55 | 1163 | | | | |
| **Species Richness** | 20 | 14 | 8 | 9 | 7 | 14 | 8 | 11 | 8 | 13 | 12 | 11 | | | | | |
| **Diversity (Shannon H′)** | 1.760 | 2.065 | 1.482 | 1.821 | 1.322 | 1.545 | 1.072 | 1.500 | 1.122 | 1.591 | 1.781 | 2.073 | | | | | |
| **Dominance (Berger–Parker)** | 0.399 | 0.324 | 0.359 | 0.320 | 0.552 | 0.521 | 0.715 | 0.560 | 0.671 | 0.451 | 0.400 | 0.382 | | | | | |

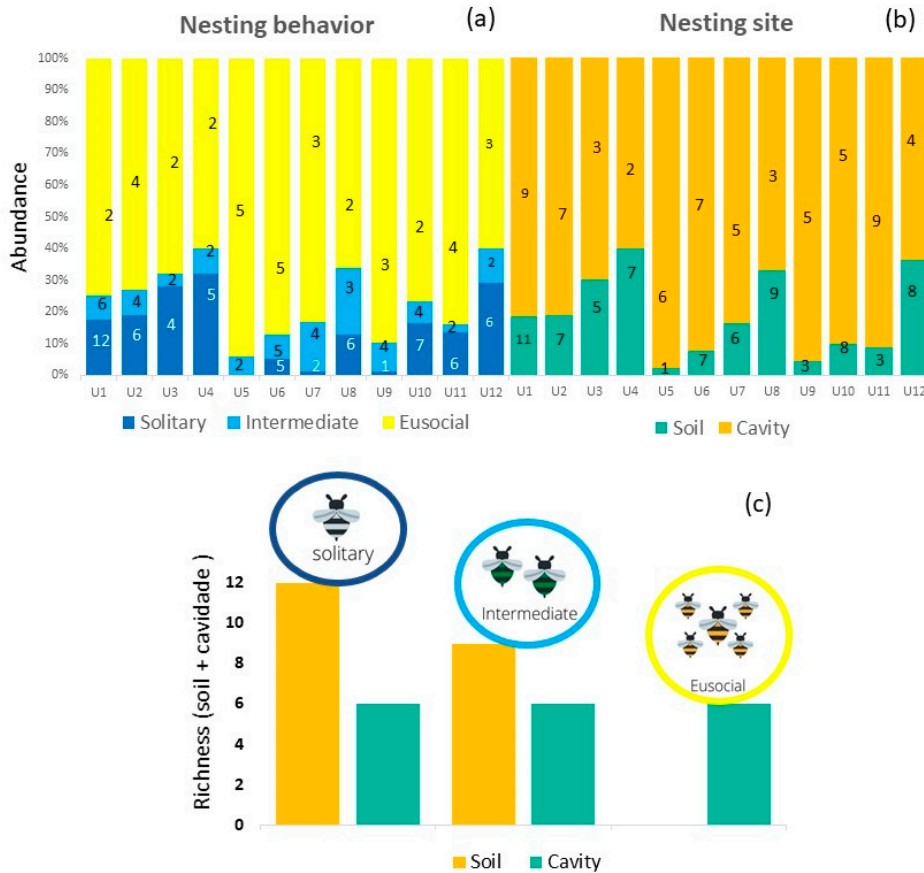

**Figure 2.** Relative abundance and richness (numbers inside bars) in each UGA: (**a**) nesting behavior, (**b**) nesting site, (**c**) species richness of soil- and cavity-nesting bees considering the different social groups.

### 3.2. Environmental Influence

Six studied UGAs presented more than 50% of their inside area as paved, and only one of the twelve UGAs had less than 50% of the surrounding paved area. The average of flower coverage in each UGA varied between 6.57 m$^2$ (U4) and 82.52 m$^2$ (U7). The most significant number of trees was registered at the U11 and the highest DBH (median = 63) at the U7 area. Three of the twelve areas studied did not have buildings higher than three floors (Table 2).

In the 12 UGAs, the bees visited 84 plant species belonging to 39 families. The average of flowering plant families among areas over the study period was 10 (ranging from 6 to 14). The families found the most in the studied UGAs were Asteraceae (10 of the 12 areas), Fabaceae (8) and Verbenaceae (8). The total plant richness per UGA ranged between 7 and 18 species. The sampling area with the highest number of plant richness was U11. The plant species with the highest frequency of occurrence in the areas were *Emilia sonchifolia* (83.3% of the areas), *Duranta erecta* (67.7%), *Tridax procumbens* (58.3%) and *Ixora coccinea* (50%). Nectar and pollen were primary floral resources for pollinators in 62 plant species, while 8 species offered only nectar, 6 species only pollen and 1 specie offered pollen and flower oil. Of the sampled bees, 55% come from flowers of plant species considered native to Brazil, and this total was mainly due to visits by Augochlorini (67% of all individuals in this tribe) and Meliponini (59%). In contrast, most bees from the tribes Apini (a species with 55% of the total number of individuals) and Halictini (a species with 58%) relate to 4 exotic plant species.

**Table 2.** Environmental conditions of 12 urban green areas studied in Campos dos Goytacazes/RJ. Inside UGA: percentage of paving (PV), number of trees (NT), diameter at breast height (DBH), plant richness (PR), flower coverage in m$^2$ (FC). Surrounding conditions measured inside a buffer of 500 m from UGA center: surrounding paved area percentage (SPV) and the number of buildings with more than three floors (NB).

| | Urban Green Areas | | | | | | | | | | | |
|---|---|---|---|---|---|---|---|---|---|---|---|---|
| | **U1** | **U2** | **U3** | **U4** | **U5** | **U6** | **U7** | **U8** | **U9** | **U10** | **U11** | **U12** |
| **PV** | 61.3 | 92.5 | 50.4 | 37.3 | 35.9 | 27.8 | 0 | 63.0 | 90.3 | 19.4 | 8.7 | 70.9 |
| **NT** | 34 | 36 | 33 | 22 | 46 | 78 | 22 | 33 | 15 | 63 | 369 | 17 |
| **DBH** | 23 | 27 | 16 | 22 | 26 | 35 | 63 | 30 | 53 | 22 | 32 | 32 |
| **PR** | 13 | 11 | 12 | 6 | 13 | 14 | 17 | 9 | 10 | 14 | 18 | 14 |
| **FC** | 60.8 | 47.5 | 10.6 | 6.5 | 42.6 | 33.7 | 82.5 | 22 | 22.7 | 26.7 | 21.3 | 11.7 |
| **SPV** | 63 | 96 | 87 | 71 | 92 | 97 | 45 | 51 | 72 | 84 | 75 | 89 |
| **NB** | 12 | 0 | 11 | 14 | 60 | 37 | 4 | 0 | 8 | 0 | 7 | 10 |

Trees' flowers received visits from 47% of the bee species, followed by herbaceous (30%) and shrubby plants (23%). Principal component analysis (PCA) showed that solitary bee correlated with herbaceous plants. Specialists, generalists, cavity and intermediate level of sociality were correlated with plants with shrubby habit, while bees from the eusocial showed correlation with trees (Figure 3).

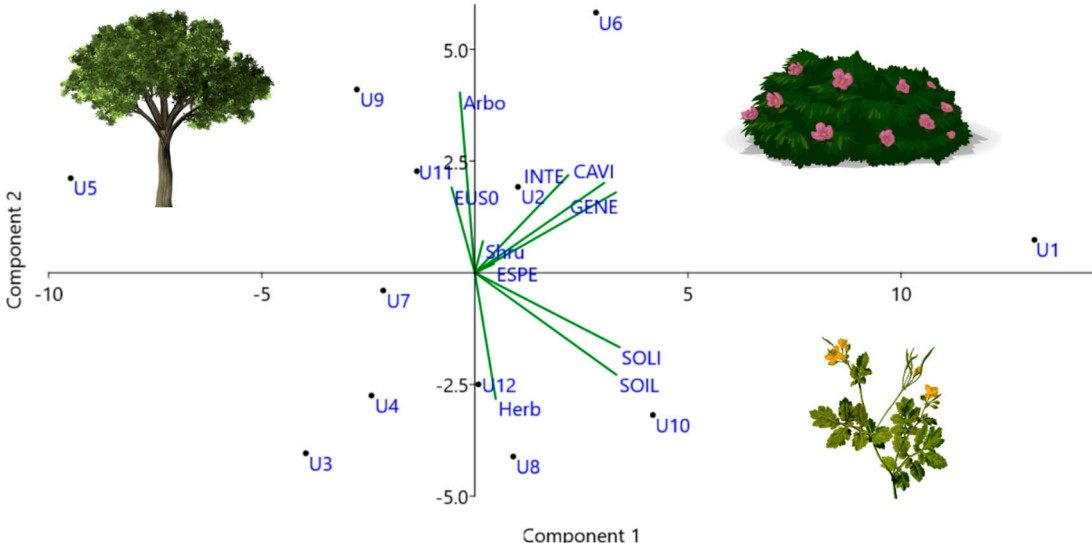

**Figure 3.** Principal Component Analysis (PCA) plot of correlation between bee guilds: SOIL (soil), CAVI (cavity), SPEC (specialist), GENE (generalist), SOLI (solitary), INTE (intermediate), EUSO (eusocial), and plant habits: Arbo (arboreal), Shru (shrubby) and Herb (herbaceous), U1 to U12 represent the urban green areas studied in Campos dos Goytacazes, Brazil.

Flower coverage was the predictor variable that more influenced the guilds analyzed in this study. The number of buildings with more than three floors explained the abundance of solitary and soil bees. The DBH negatively impacted the abundance of specialist and solitary bees. The paving percentage did not influence the bees' responses, and the area and number of trees were excluded from analyses because they presented collinearity (Table 3).

**Table 3.** Parameter estimates of a generalized linear model, explaining abundance and richness of bee guilds in urban green area (UGA) in Campos dos Goytacazes, RJ, Brazil. Inside UGA: flower cover in m$^2$ (FC), diameter at breast height (DBH), plant richness (PR), surrounding percentage of paving (SPV) and the number of buildings with more than three floors (NB).

| Variables | Model | FC | DBH | PR | SPV | NB | Intercept | df | AICc | ΔAICc | Weight | AdjR$^2$ |
|---|---|---|---|---|---|---|---|---|---|---|---|---|
| **Abundance** | | | | | | | | | | | | |
| Eusocial | 2 | | 0.029 | | | | 3.186 | 3 | 123.0 | 0.00 | 0.405 | 0.396 |
| Intermediate | 11 | | | | −0.289 | | 4.301 | 3 | 76.8 | 0.00 | 0.318 | 0.477 |
| Solitary | 19 | | −0.052 | | | −0.038 | 4.350 | 4 | 82.0 | 0.00 | 0.329 | 0.069 |
| Soil | 17 | | | | | −0.032 | 3.102 | 3 | 88.4 | 0.00 | 0.355 | 0.436 |
| Cavity | 2 | 0.015 | | | | | 3.830 | 3 | 127.6 | 0.00 | 0.276 | 0.318 |
| Specialist | 4 | 1.447 | −1.521 | | | | −50.88 | 4 | 17.8 | 0.00 | 0.217 | 0.993 |
| Generalist | 2 | 0.015 | | | | | 4.096 | 3 | 127.1 | 0.00 | 0.285 | 0.345 |
| **Richness** | | | | | | | | | | | | |
| Eusocial | 17 | | | | | 0.012 | 0.937 | 3 | 47.3 | 1.77 | 0.134 | 0.151 |
| Intermediate | 3 | 0.101 | | | | | 0.851 | 3 | 49.2 | 1.51 | 0.168 | 0.169 |
| Solitary | 2 | | −0.029 | | | | 2.486 | 3 | 65.8 | 0.00 | 0.179 | 0.264 |
| Soil | 17 | | | | | −0.015 | 2.034 | 3 | 65.7 | 0.41 | 0.215 | 0.239 |
| Cavity | 129 | | | 0.079 | | | 0.661 | 3 | 57.6 | 0.00 | 0.143 | 0.282 |
| Specialist | 2 | 0.047 | | | | | −46.38 | 3 | 14.6 | 2.27 | 0.095 | 0.250 |
| Generalist | 2 | 0.002 | | | | | 2.311 | 3 | 66.7 | 3.23 | 0.083 | 0.035 |

## 4. Discussion

The study data suggest that despite being a medium-sized city, the study area is at a high level of urbanization, with percentages of paving above 50% [16] and urban densification. Large paved areas have already replaced the natural areas at this stage, where only low flower covers and few potential nesting places for bees remain, making these areas harmful to the pollinator community [4]. This condition highlights the importance of urban green areas as bees' refuge, functioning as a shelter for different bee species. The lack of previous data does not allow comparisons of community parameters over time. However, the comparison between areas enables us to realize variations in bee composition. The presence of robust bees such as *X. frontalis, X. nigrocincta* and *E. nigrita* in UGA (U2, U5, U6), with a high percentile of paved surrounding (over 90%), corroborates the idea that UGAs can provide resources in scarcity situations. These bees have a long flight range and can cross large distances in the urban matrix searching for food.

Results indicated that the different guilds of bees diverged in responses to the conditions found in urban green areas and their surroundings. The structure of the bee community showed a composition mainly of eusocial and generalist bees, typical in open and highly modified environments. Other studies in Brazil have shown the tendency for eusocial bees to be more abundant in urban areas [32–35]. The fact that the colonies of eusocial species have thousands of individuals and are active throughout the year explains these results, allowing to take advantage of resources from plant species with different phenological cycles [15]. In addition, eusocial bees recruit and can collect a large number of resources quickly. Therefore, in places with little diversity of resources, these eusocial bees usually benefit compared to solitary bees.

In 50% of UGAs, the most abundant species was *Apis mellifera* (32% of individuals sampled). The more remarkable plasticity of *A. mellifera* favors the increase of its abundance in urban areas and the competitive pressure on native species, which can lead to the homogenization of the structure of the bee community in these areas [36] through the replacement of native bee species by this exotic one [37]. Although several human activities promote biotic homogenization, urbanization is the one that most favors this process [38,39]. Cities are made to attend to human needs, and therefore have a uniform nature, repeated throughout the world, with buildings, roads and houses. The construction of cities destroys the habitat of many native species, but on the other hand, creates habitats for other species, such as the exotic *A. mellifera* that can use cavities in human constructions to build nests. Consequently, more vulnerable species tend to disappear, decreasing the community species richness [38,39].

The similarity of 67% between U7 and U8 draws attention, as they are about 5 km apart. Their surrounding conditions are similar, with lower percentages of surrounding paving (41% and 51%, respectively) and low numbers of buildings with more than three floors ($n = 4$ and $n = 0$). Both are in the city borderline connected with vegetated areas. On the other hand, U7 and U4—the less similar areas, are only 700 m apart, and U4 is surrounded by a high percentage of paved areas (71%) and buildings ($n = 14$) and had low flower coverage (6%). These environmental conditions can compromise the movement of bees between areas and influence the composition of the bee's community since it responds in small scales to the characteristics of the environment associated with the supply of resources and nesting site [40].

The distribution of bees between native and exotic plant species did not explain the abundance and richness of the different guilds between the areas. The average flower coverage was 32%, and considering that this variable can measure the available resource, it is possible that the bees are using exotic plants as an alternative to obtain resources. This behavior is present in bees studied in an environmental gradient in the United States, aiming to examine the importance of exotic plants for native bee species [41]. In environments with anthropogenic disturbances, bees used many exotic species, and the authors found a correlation between visits to exotic plants and the low abundance of plants in the environment. This study corroborates this idea because 23% of bees sampled in our study were in shrub species, with 16% sampled in three species: *Duranta erecta*, *Ixora coccinea* and *Ruellia simplex*. All three are exotic species widely used in urban afforestation in the region. Resistant to heat and lack of water, they bloom year-round and have often been one of the few resource sources for pollinators in the UGAs.

Plant habit appears as a determinant in the distribution of bee guilds. The work highlights the importance of herbaceous plants for maintaining the solitary bees and those that nest in the soil. Among them 63% were collected in herbaceous plant species (25% solitary, 38% soil-nesting bees). Herbaceous plants were considered an essential source of resources and attractive to a large number of visitors in a study carried out in a restinga (coastal sand area) conservation unit in the state of Rio de Janeiro (Brazil) [42]. According to the authors, naturally born herbaceous/pioneers were associated with higher richness and abundance of floral visitors in restoration areas in restinga remnants. Different herbaceous plants offer various resources that can benefit wild bees when supplying their brood cells [43]. As herbaceous plants are born spontaneously in areas of exposed soil, their presence in UGAs represents potential resources of foraging and nesting places for bees that nest in the soil.

The abundance of eusocial bee appears associated with plant species of arboreal habit. Eusocial bees forage in groups and demand a greater abundance of flowers. Among the eusocial species are the stingless bees considered the primary pollinators of tropical trees with "mass" flowering [44]. Furthermore, eusocial species nest in pre-existing cavities in tree trunks, increasing the importance of tree species to native eusocial species.

Among the guilds that demonstrate greater vulnerability to the urban environment are the specialist bees. In this study, we found 1% of specialist bees in the community, similar to that described by other studies in the urban area [16]. The decline of specialist bees can be directly related to the loss of the host plant, as described in a study that found the decline of specialist bees associated with the decline of Fabaceae plants caused by the management of agricultural areas and the removal of native vegetation [45].

Analyzing the abundance of bees that nest in the soil, we found a sensitivity related to the number of buildings with more than three floors. Of the 39 species of bees sampled in this study, 16% nest in the soil. As the number of buildings increases, there is less abundance of bees from this guild in the UGAs. These high buildings are usually in large areas of paving surface that limit nesting sites as a consequence of the replacement of exposed soil areas by paved areas and the removal of small bushes or spontaneous vegetation that provide food resources [35]. However, for bees that nest in the soil, the

availability of nesting sites is a decisive factor in the establishment and growth of their populations [46].

Considering that 60% of solitary bees are also soil-nesting bees, they are also influenced by a variable "number of buildings with more than three floors" on richness and abundance. A lower abundance of solitary bees appeared overall, with only 11% of the captured bees belonging to solitary species. The urban environment is composed of mosaics of vegetation and buildings that probably prevent or hinder the circulation of bees with a small flight radius, such as species of Augochlorini that we found in this study. Few individuals in this group reach long flight ranges, which increases the dependence of solitary bees on the resources available near the nests. The distance from the resource determines the ability to maintain the species [47]. According to the authors, the number of descendants generated by the species *Osmia lignaria* was more significant when the nest was close to areas with a good supply of resources. We can expect a similar situation for solitary bees in the studied areas.

The results of the solitary, soil and intermediate bees influenced by the "number of buildings with more than three floors" and by the surrounding pavement demonstrate the importance of creating green corridors that increase the possibility of bees' movement, avoiding isolation. According to a recent study [48], it is not urban environments that negatively influence the response of bee communities, but the way cities are organized act as filters that limit the presence of some species. Our results corroborate this affirmation as they indicated that the green areas and surroundings do not consider the necessities of different bee species.

## 5. Conclusions

We suggest that decisions about management of urban green areas must be transdisciplinary in decision making and planning. We need engineers and architects that comprehend the importance of including green areas in their projects, and of keeping native vegetation. We need public managers that consider the creation of parks and squares a priority in the city. We must also make residents understand the importance of trees and vegetation for a healthy city and to not exchange vegetation for paving. A bee conservation program in urban areas should include the management of plants that provide resources for bees and other pollinators in abundance all year. Creating and maintaining of urban parks with more vegetation cover is important to maintain the milder climate and increase the diversity of essential animals, such as pollinators and seed dispersers, in the urban area. This vision constitutes two of the UN Sustainable Development Goals (SDGs): item 15, which deals with terrestrial life and aims to protect, recover and promote the sustainable use of terrestrial ecosystems, sustainably manage forests, combat desertification, stop and reverse the degradation of the Earth and stop the loss of biodiversity, and item 11, which deals with cities and sustainable communities to make urban spaces and human settlements inclusive, safe, resilient and sustainable. This work considers that the planned urban policies, considering other organisms besides human beings, can lessen the negative impacts of rapid urbanization. A new way of looking at the city is the one that considers man as part of the ecosystem and that by conserving other species will be conserving its quality of life in the urban environment.

**Author Contributions:** M.C.G. and S.G.A. conceived the ideas and designed the methodology; S.G.A. collected the data; M.C.G. and S.G.A. analyzed the data; S.G.A. and M.C.G. wrote the paper. All authors contributed critically to the drafts. All authors have read and agreed to the published version of the manuscript.

**Funding:** This research was funded by CNPq (303894/2018-0) and FAPERJ (203.321/2017), and SGA was funded by Coordenação de Aperfeiçoamento de Pessoal de Nível Superior-Brasil (CAPES)—Finance Code 001.

**Institutional Review Board Statement:** Fieldwork and laboratory procedures were in compliance with CEUA—Ethical Committee for the use of animals' law n° 11.794/08, invertebrates don't require authorization.

**Informed Consent Statement:** Not applicable.

**Data Availability Statement:** The data available in this study are available on request from the corresponding author. The data are not publicly available due to needing further use.

**Acknowledgments:** We thank the graduation Program of Ecology and Natural Resources from Universidade Estadual do Norte Fluminense Darcy Ribeiro. We also thank Campos dos Goytacazes city hall for the permission to study in public areas, Janie Mendes Jasmim for the help in plant identification, Caique Barcelos da Silva for field help, Camila Priante for making the location map and Lucas Carneiro e Lazaro Carneiro for statistical analysis help. This study was financed in part by the Coordenação de Aperfeiçoamento de Pessoal de Nível Superior-Brasil (CAPES)—Finance Code 001. M.C.G. thanks CNPq (303894/2018-0) and FAPERJ (203.321/2017) for financial support.

**Conflicts of Interest:** The authors declare no conflict of interest.

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
