# Peer review of "Bee Guilds’ Responses to Urbanization in Neotropics: A Case Studyâ€"

_diversity, doi:10.3390/d13080365_

Round 1
Reviewer 1 Report
This is a great manuscript which should be published. The content is very topical and will solicit much interest from Diversity's readership. The methods are well described and the statistical analysis looks appropriate.
One potential reservation to consider is that only one urban area was sampled (of course due to the amount of effort involved), and the findings should not be extrapolated too much to other similar urban areas. It is safest to refer to this urban sampling as a case study. And of course more such work is needed, yet another reason to publish this work. I am happy with the amount of study details given in the methods and it will be a good guide for future researchers that want to carry out similar research. So to summarize this is not a major concern.
In general the manuscript is free of errors, but the Discussion needs careful checking as it has a number of minor errors (shown below)
Minor comments:
line 137: dispersal is more appropriate than circulation
line 282: is there any beekeeping done in this urban area? And if so, does this vary between your 12 sites?
line 287: "Cities are..." instead of "The city is..."
line 306: " a proxy of disponible resource " not sure what this means?
line 317: Should "Habit" not be Habitat?
line 323: full stop missing
Line 343: more than three pavements or floors?
line 364-5: I think "movement" instead of "circulation" is a better description.
References in line 491 and 506: wrong font
Author Response
Reviewer #1:
Thank you for the revision. Thank you for the positive comments about our manuscript and all other suggestions. All the suggestions were taken into account to improve the quality of the document.
Reviewer: This is a great manuscript which should be published. The content is very topical and will solicit much interest from Diversity's readership. The methods are well described and the statistical analysis looks appropriate. One potential reservation to consider is that only one urban area was sampled (of course due to the amount of effort involved), and the findings should not be extrapolated too much to other similar urban areas. It is safest to refer to this urban sampling as a case study. And of course, more such work is needed, yet another reason to publish this work. I am happy with the amount of study details given in the methods and it will be a good guide for future researchers that want to carry out similar research. So, to summarize this is not a major concern.
Response: We inserted “ study case” in tittle answering you suggestion
Reviewer: In general, the manuscript is free of errors, but the Discussion needs careful checking as it has a number of minor errors (shown below). Minor comments: line 137: dispersal is more appropriate than circulation
Response: Ok (line 137)
line 287: "Cities are..." instead of "The city is..."
Response: OK (line 275)
Reviewer: line 282: is there any beekeeping done in this urban area? And if so, does this vary between your 12 sites?
Response: No, there is no beekeeping in urban area.
Reviewer: line 306: " a proxy of disponible resource " not sure what this means?
Response: Change to : “ …can measure available resource…” (line 293)
Reviewer: line 317: Should "Habit" not be Habitat?
Response: In this case we were talking about plant habit, so now we included the word “plant” before habit to be clearer. (line 304)
Reviewer: line 323: full stop missing
Response: Included (line 310)
Reviewer: Line 343: more than three pavements or floors?
Response: Changed “to floors” (line 329)
Reviewer: line 364-5: I think "movement" instead of "circulation" is a better description.
Response: Included (350)
Reviewer: References in line 491 and 506: wrong font
Response : Ok (lines 480 and 495)

Reviewer 2 Report
raw 15 – DPH is not specified
raw 30 – „Usually, the urbanization process is not planned...“
This should be more specified geographically since the urbanization is carefully planned in Western Europe for example and surely also in other parts of the world.
raw 41 - „...or even exposed.“
.. not clear expression (?)
raw 52 – „...there is no application of pesticides...“
The pesticides are used also in cities (private gardens, lawns, flower belts...), even when less intensively than in agricultural landscape, see:
https://doi.org/10.3390/su12229427
It may be another explanatory factor to be taken into account as concerns the differences in bee communities between study sites (?).
Reference to Figure 1 is missing in the text
raw 234 – „In the studied areas.“
... sentence without sense (?)
Author Response
Reviewer # 2
Thank you for the suggestions, all were taken into account to improve the quality of the document. Thank you for the revision.
Reviewer: raw 15 – DPH is not specified
Response: Ok, included (line 15-16)
Reviewer: raw 30 – „Usually, the urbanization process is not planned... “This should be more specified geographically since the urbanization is carefully planned in Western Europe for example and surely also in other parts of the world.
Response: Included “… in tropical and developing countries.” (lines 32,33)
Reviewer raw 41 – “...or even exposed. “.. not clear expression (?)
Response: Included “… exposed, out of cavities” (line 43)
Reviewer raw 52 – „...there is no application of pesticides... “
The pesticides are used also in cities (private gardens, lawns, flower belts...), even when less intensively than in agricultural landscape, see: https://doi.org/10.3390/su12229427. It may be another explanatory factor to be taken into account as concerns the differences in bee communities between study sites (?).
Response: Inserted “lower” application of pesticides (line 55)
Yes, in this study we found that location, like green areas in borderline, influenced in community similarity. Also, flower cover, DBH and number of buildings with more than three floors influenced the responses of bee guilds in different green areas.
Reviewer: Reference to Figure 1 is missing in the text
Response: Ok, included (line 99)
Reviewer: raw 234 – „In the studied areas. “... sentence without sense (?)
Response: Thank you, it was really wrong, some sentence lost, probably we forgot to exclude. We made it now.
